# Examination of the Responsiveness of the Great Recess Framework—Observational Tool

**DOI:** 10.3390/ijerph17010225

**Published:** 2019-12-28

**Authors:** William V. Massey, Megan B. Stellino, Laura Hayden, Janelle Thalken

**Affiliations:** 1College of Public Health and Human Sciences, School of Biological and Population Health Sciences, Kinesiology Program, Oregon State University, Corvallis, OR 97331, USA; thalkenj@oregonstate.edu; 2School of Sport and Exercise Science, University of Northern Colorado, Greeley, CO 80639, USA; megan.stellino@unco.edu; 3College of Education and Human Development, Counseling and School Psychology, University of Massachusetts at Boston, Boston, MA 02125, USA; Laura.Hayden@umb.edu

**Keywords:** recess, physical activity, school health, measurement

## Abstract

The purpose of this study was to test the responsiveness of the great recess framework-observational tool (GRF-OT) to detect changes in recess quality. GRF-OT data were collected at two time points (fall 2017 and spring 2018) in four geographically distinct regions of the United States. Following recommendations by Massey et al. (2018), a three-day average of recess observations was used for each data point. Data analysis was conducted on nine schools contracted to receive services from Playworks, a national non-profit organization specializing in recess implementation, for the first time; eight schools with returning Playworks services (i.e., multiple years of service) and five schools with no intervention services. Analysis of the change in GRF-OT scores from fall to spring revealed a large effect for first-year intervention schools (g = 1.19; 95% CI 0.13, 2.25) and multi-year intervention schools (g = 0.788; 95% CI −0.204, 1.78). GRF-OT scores decreased for schools not receiving an intervention (g = −0.562; 95% CI, −2.20, 1.07). New intervention schools (odds ratio= 21.59; 95% CI 4.27, 109.15) and multi-year intervention schools (odds ratio= 7.34; 95% CI 1.50, 35.97) were more likely to meet the threshold for meaningful positive change than non-intervention schools. The results of the current study suggest that GRF-OT is a responsive tool that researchers, practitioners, and policy makers can use to measure and assess changes in the quality of the recess environment.

## 1. Introduction

School-based recess continues to draw attention as an important opportunity for both physical activity (PA) and child development outcomes. Given this, validated outcome assessments are necessary to understand what happens at recess, as well as if intervention efforts to improve the recess climate are successful. However, PA remains the primary outcome associated with recess, while often ignoring the environmental, social, and behavioral aspects of recess that might contribute to child development outcomes [1]. Previously, researchers have created observational instruments to assess levels of physical activity and social behavior at recess [2,3] however these instruments are reliant on time sampling techniques that observe one child at a time and likely miss out on larger environmental influences. More recently, Grady-Dominguez and colleagues utilized a Rasch analysis to validate an instrument that examined play sophistication during recess, however this instrument was developed specific for children with autism and intellectual disabilities [4]. To date, the only known observational instrument of recess that examines what happens at recess overall, and contains multiple constructs (e.g., environmental, social, behavioral) is the great recess framework-observational tool (GRF-OT) developed by Massey and colleagues [5].

The GRF-OT contains 17 items that each describe in short detail critical aspects of the recess environment within four sub-domains: safety and structure, student behaviors, adult engagement and supervision, and transitions. Items related to safety and structure focus on the built environment, the availability of equipment and materials to support play, and the use of equipment and materials during recess. Items related to adult engagement and supervision assess the number of adults on the playground, as well as whether or not adults are supporting and engaging with students during recess. Student behavior items assess various social behaviors such as whether or not students are able to initiate and sustain their own play, solve conflicts, and the level of physical and verbal altercations. Finally, items related to transitions examine the transitions from the classroom to recess and from recess to the classroom. Response formats for all items are scores on a 1 (low quality) to 4 (high quality) rating scale, with higher scores representing higher quality markers during recess. Previous researchers [5] have established the content validity, factor structure and measurement validity, inter-rater reliability, and test-retest reliability of the GRF-OT. In doing so, it was recommended that a three-day average be used to assess recess quality, and a change score of 4.62 on the GRF-OT was needed to detect change beyond the variability expected in recess observations. Subsequent data in independent samples supported the validity of the GRF-OT, as recess sessions scoring higher on the GRF-OT also had higher levels of student engagement in play and games at recess [6]. However, these findings have been from cross-sectional data and do not support the use of the GRF-OT to track change over time. Determination of whether or not the GRF-OT is sensitive enough to detect change is a necessary step in substantiation of the monitoring and evaluation of recess. As such, the purpose of the current study was to test the responsiveness of the GRF-OT to detect changes in recess quality across a schoolyear.

To determine the responsiveness of the GRF-OT, the study team collaborated with Playworks (www.playworks.org). Playworks is a non-profit organization that seeks to provide opportunities for developmentally appropriate recess for children in low-income school districts. The Playworks model of practice includes key constructs of the GRF-OT that would reasonably result in observed changes in recess quality, thereby making it an appropriate intervention to test the ability of the GRF-OT to detect change. Playworks has trained adults (i.e., “coaches”) in schools serving low-income communities who provide opportunities for students to engage in play throughout the day. By placing these adults on the playground, and training teachers to engage in recess facilitation, changes should be seen in the GRF-OT adult supervision and engagement items. Previous research has shown that total adult-student interactions, and positive adult-student interactions increase overtime with Playworks, while punishment from adults on the playground decreases over time [7]. During recesses, Playworks coaches provide equipment, lead and organize games and activities that empower students and school staff to take control of the recess environment in a positive manner. As part of the program, Playworks’ coaches also work with classroom teachers and their students during class game times, which allow for physical activity and help to establish rules, boundaries, and expectations that translate to the recess context. These changes should result in positive increases in GRF-OT items related to boundaries, availability of equipment, games, and participation in active play. Results from previous randomized controlled trials have shown higher levels of physical activity for girls [8] and minority students [9]. The Playworks program also provides student leadership opportunities through a Junior Coach leadership program where students are recruited and trained to become leaders amongst their peers. These changes should affect various student behaviors on the playground. Results of previous RCTs have shown lower levels of bullying and less transition difficulty at recess following Playworks program implementation [10,11,12]. Previous research has shown overall increases in recess quality [13] and improvements in problem solving skills over time with the Playworks program [14]. Based on these data, it was assumed that schools receiving Playworks interventions would experience positive changes in recess quality. The purpose of the study was not to asses if the Playworks program changed recess or to evaluate the program, but rather, if the GRF-OT can detect change overtime. It was hypothesized that higher positive change scores on the GRF-OT would be observed in first-year Playworks intervention schools compared to returning intervention schools and non-intervention schools. It was also hypothesized that higher positive change scores on the GRF-OT would be observed in returning intervention schools when compared to non-intervention schools.

## 2. Materials and Methods

### 2.1. Method

Observational data were collected at 26 schools representing four distinct regions of the United States (Pacific Northwest (PNW), Rocky Mountains (RM), Midwest (MW), Northeast (NE)). Across the 26 schools, a standard structure to recess did not exist; rather recess schedules varied as a function of individual school schedules. On average, approximately four recess blocks were scheduled per school, per day, resulting in data being collected at 116 unique recess blocks (i.e., different children within each block). Table 1 provides school-level demographic information for each data collection site.

### 2.2. Measures

The GRF-OT is an observational measure that is used for live data collection during recess. Previous research has shown support for the factorial validity of a four-factor model for the GRF-OT [5]. Specifically, safety and structure of the playground, adult supervision and engagement, student behaviors, and transitions have all emerged as unique constructs within the tool. These constructs are measured through a series of questions that pertain to the context of recess and placed on a 4-point scale in which lower scores reflect lower quality scores for each item. The GRF-OT provides an overall score for recess quality on 17-items that represent the four latent constructs described above. Data collection is completed by a trained data assessor who (a) conducts an audit of the recess environment prior to recess, (b) observes live recess sessions while completing process notes for each scored item, and (c) provides an overall score for each scored item. The range of possible scores on the GRF-OT is 17 (i.e., “1” for each item) to 68 (i.e., “4” for each item). Previous data has supported the inter-rater reliability, and the test-retest stability of the GRF-OT [5]. The GRF-OT items and scoring procedures are freely available at www.greatrecessframework.org.

### 2.3. Procedures

This study was deemed exempt by the institutional review board providing study oversight. Data assessors were graduate students who were blinded to the study purpose and hypotheses. On two occasions, the study PI had to assist with data collection, and thus blinded, was compromised. In each case, the study PI was blinded to previous observations. Prior to data collection, all data assessors completed an online training module that documented the observational protocol and scoring system. This was followed by a one-hour workshop led by the first author in which the scoring protocol was discussed in greater depth, with various examples and scenarios provided. Training workshops were conducted face-to-face in two locations, and via video conferencing in two locations. In two study locations (one face-to-face training; one video conference training), data assessors were accompanied by expert assessors during baseline data collection to ensure adherence to scoring criteria. As our previous research has documented inter-rated reliability on the GRF-OT for trained graduate students, inter-rater reliability data were not collected in the current study. In all locations, process notes were documented by the data assessor and used to audit scoring on the GRF-OT. Data collection was completed during the first month of the academic school year, and again in the spring of the same academic year at all schools. At each time point, data assessors completed three observations of recess within a seven-day time period. The three-day average was used as the overall recess quality score in data analysis. Because schools were paying for Playworks services, intervention programming was not delayed for data collection purposes. Thus, Playworks interventions were in place during baseline data collection. Data in the current study show significant differences in baseline scores between new Playworks schools (52.13) and non-intervention schools (47.61; *p* = 0.001) which may suggest baseline scores were inflated in the new Playworks condition. Inflated baseline scores would make it more difficult to track change over time.

Of the 26 schools that agreed to participate, 13 schools were receiving the Playworks intervention for the first time, eight schools were receiving the Playworks intervention for at least the second consecutive year, and 5 schools had requested Playworks intervention but were not receiving the intervention during the year data were collected. Following study completion, but prior to data analysis, schools were contacted to determine outside factors that may have impacted intervention effectiveness, and thus rendered them ineffective data points for determining the responsiveness to change for the GRF-OT. As a result of this, two schools were eliminated from analysis due to a change in school leadership during the data collection period (School 1, School 25). As Playworks model of intervention is predicated on investment from, and behavior change associated with school staff, a change in leadership and school priorities in-between data collection periods was seen as likely to alter the investment in recess, and thus were considered unreliable to determine how well the tool itself is able to detect change. An additional two schools were eliminated as Playworks staff noted difficulties in program implementation (School 12 and School 18). While excluded from statistical analyses, we have presented the descriptive data from these four schools in Table 1 and Table 2. Given this, the final sample for analysis included 22 schools and 97 unique recess blocks. Nine schools were receiving Playworks services for the first time (36 recess blocks), eight schools were in at least their second year of Playworks programming (39 recess blocks), and five were school receiving no recess intervention (21 recess blocks).

### 2.4. Data Analysis

Data were collected to measure the responsiveness of the GRF-OT to detect change across an academic school year. To test the hypothesis that higher positive change scores on the GRF-OT would be observed in schools receiving Playworks for the first time, the magnitude of the beginning- to end-of-year change scores were calculated using Hedges’ *g* and 95% confidence intervals for each respective intervention group. Hedges’ *g* calculations were interpreted as small (≈0.20), medium (≈0.50), and large (≈0.80 or larger). Next, each individual recess block was classified as meeting or failing to meet the GRF-OT minimally detectable change (MDC) score of 4.62. A logistic regression was then conducted to examine if intervention condition was predictive of meeting the MDC. Finally, separate 3 (Group) × 2 (Time) Factorial Analysis of Covariance (ANCOVA) were conducted to examine different patterns of change across time for each group on the total scale and each sub-scale. GRF-OT score was entered as the dependent variable, and school and geographic region were dummy coded and entered as covariates. Effect sizes are expressed as partial eta squared, and interpreted as small (≈0.02), medium (≈0.06), and large (≈0.14 or larger).

## 3. Results

At the beginning of the school year (Time 1), the average GRF-OT score was 52.83 (3.11/4 per item) and ranged from 41 (2.41/4 per item) to 65.33 (3.84/4 per item). GRF-OT scores were lowest at non-intervention schools (M_score_ = 47.61, SD = 5.40), followed by new Playworks schools (M_score_ = 52.12, SD = 4.54), and returning Playworks schools (M_score_ = 56.29, SD = 5.88). In the spring (Time 2), the average GRF-OT score was 55.66 (3.27/4 per item) and ranged from 33.33 (1.96/4 per item) to 67.00 (3.94/4 per item). GRF-OT scores were lowest at non-intervention schools (M_score_ = 44.52, SD = 5.40), followed by new Playworks schools (M_score_ = 57.63, SD = 4.60), and returning Playworks schools (M_score_ = 59.85, SD = 2.33). To assess the magnitude of change from Time 1 to spring, Time 2, effect sizes were calculated using Hedge’s *g* for each group respectively (new Playworks, returning Playworks, no intervention). For those new to Playworks, there was a large magnitude of change detected by the GRF-OT (*g* = 1.19; 95% CI 0.13, 2.25). Similarly, there was a large magnitude of change detected by the GRF-OT (*g* = 0.788; 95% CI −0.204, 1.78) for returning Playworks schools. For those schools not receiving Playworks programming, there was a negative (i.e., GRF-OT scores decreased), moderate effect (*g* = −0.562; 95% CI, −2.20, 1.07).

New Playworks schools were significantly more likely to meet the MDC threshold than non-intervention schools (*p* < 0.001; Odds Ratio = 21.59; 95% CI 4.27, 109.15). Returning Playworks schools were also significantly more likely to meet the MDC threshold than non-intervention schools (*p* = 0.014; Odds Ratio = 7.34; 95% CI 1.50, 35.95). Average change scores for each school can be seen in Table 1. There was a significant difference in baseline scores between intervention schools that met versus did not meet the MDC threshold (*p* < 0.001). Schools that did not meet the MDC threshold had higher baseline recess quality scores (57.97) as compared to schools that met the MDC (51.40).

Finally, results of the three (new intervention, returning intervention, non-intervention) × 2 (beginning of year, end of year) ANCOVA showed a significant interaction effect (*p* < 0.001; partial eta squared = 0.328), indicating a significantly different pattern of change across the three groups. As shown in both Table 2 and Figure 1, recess sessions with Playworks intervention increased their GRF-OT score throughout the year, whereas those with no intervention decreased their GRF-OT score. A similar pattern of results was observed for individual sub-scales of the GRF-OT. As can be seen in Figure 2, when controlling for both school, and region, a differential pattern of change was detected between the three intervention groups for *safety and structure* (*p* < 0.001; partial eta squared = 0.341), *adult engagement and supervision* (*p* = 0.072; partial eta squared = 0.056), *student behaviors* (*p* < 0.001; partial eta squared = 0.207), and *transitions* (*p* = 0.011; partial eta squared = 0.095).

## 4. Discussion

The purpose of the current study was to test the responsiveness of the GRF-OT to detect meaningful changes in recess quality across a schoolyear. There was a large magnitude of positive change detected for first year Playworks intervention schools, as well as returning Playworks intervention schools. Moreover, a moderate negative magnitude of change was detected for schools that were not receiving an intervention, suggesting that recess quality may diminish across the schoolyear. Each of the four GRF-OT sub-scales, (1) safety and structure, (2) adult engagement and supervision, (3) student behaviors, and (4) transitions were also shown to be responsive to change, as demonstrated by the differential patterns across intervention types. It should be noted that adult engagement and supervision sub-scale was not significantly different between groups when controlling for school and region, however a moderate effect was observed.

Previous research [5,6] has established the reliability and validity of the GRF-OT, however the current study makes a unique contribution to the literature in examining the responsiveness of this tool to detect change over time. While levels of PA continue to be the primary outcome measure associated with school-based recess, understanding how recess may contribute to the social development of children necessitates a more holistic means of evaluation of this context. Notably, if children are active in an environment that includes high levels of bullying and anti-social behavior, it is likely that PA alone will not contribute positively to their social development [15]. Moreover, research has shown children’s experiences of recess differ greatly, and may be effected by equity of space and equipment use [16], gender [17], adult engagement [7], and social status [18]. Thus, comprehensive and thorough evaluation of how the environment shapes various outcomes is paramount to understanding how recess contributes to the health and wellbeing of children in schools. As initiatives driven by the CDC and SHAPE America include the ongoing monitoring and assessment of recess as a necessary strategy for healthy schools, the GRF-OT is a reliable, valid and now evidence-based responsive tool that is freely available as a resource for schools to consider in this process.

## 5. Limitations and Directions for Future Research

Despite a positive contribution to the literature, the current study is not without limitations. First, aside from situations in which interventions may have been ineffective (e.g., administrative changes), assessment of perceived changes in the recess environment were not examined. Future research should consider key stakeholders’ perceptions of recess changes (e.g., students, teachers, administration) and compare these data to changes observed using the GRF-OT. Moreover, including additional timepoints, as opposed to just the beginning and end of the school year, would allow for examination of whether or not the GRF-OT can detect more subtle changes over time. Another limitation to the current study is that true baseline data for intervention schools was not possible, as Playworks begins their program at the beginning of the academic school year. Thus, at intervention schools, Time 1 data collection was in the early phases of the Playworks intervention. This could possibly explain why intervention schools had higher baseline scores than non-intervention schools and might also under-estimate the magnitude of change detected by the GRF-OT. Indeed, analysis showed those with higher baseline scores were less likely to meet the threshold of the MDC, which may indicate a ceiling effect for the GRF-OT. Finally, it should be noted that the current study assessed whether the GRF-OT could detect change within a recess intervention that provided a more structured recess environment, and it is unclear as to whether or not change would be detected in other forms of intervention (e.g., the introduction of loose parts or playground marking to encourage play).

In considering a future line of research using the GRF-OT, data examining how recess quality affects children’s physical activity, social-emotional development, classroom behavior and academic performance are important to understand the impacts of recess quality. At this time, both observational and experimental studies are needed to better understand this relationship, thereby informing policy decisions around recess.

## 6. Conclusions

As efforts to monitor and evaluate recess outcomes and contexts become increasingly important for school compliance and public policy decisions, access to an instrument that can assess the quality of the recess environment and is able to detect change in quality over time is imperative. This study adds to the literature by providing a valid, reliable, and change-sensitive measure for use in future research on school-based recess. The results of this study suggest the GRF-OT is a responsive instrument that can be used by schools, practitioners, and policy makers to assess the quality of the recess environment.

## Figures and Tables

**Figure 1 ijerph-17-00225-f001:**
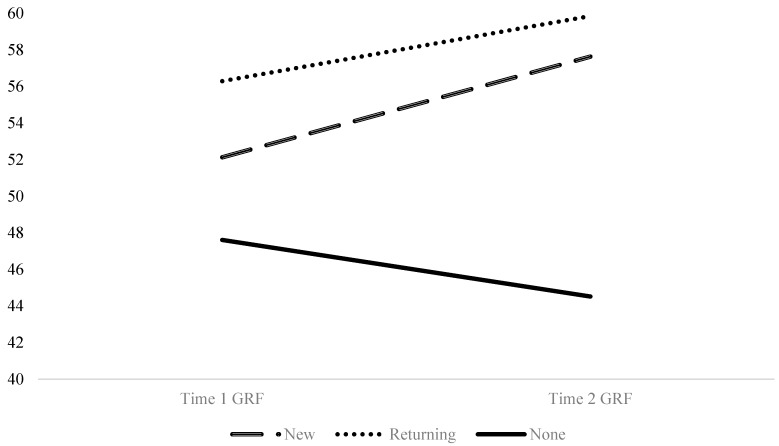
Group by time interaction of Great Recess Framework-Observational Tool (GRF-OT) scores show that recess sessions with Playworks programming increased their GRF-OT score throughout the year, while those with no programming decreased their GRF-OT score.

**Figure 2 ijerph-17-00225-f002:**
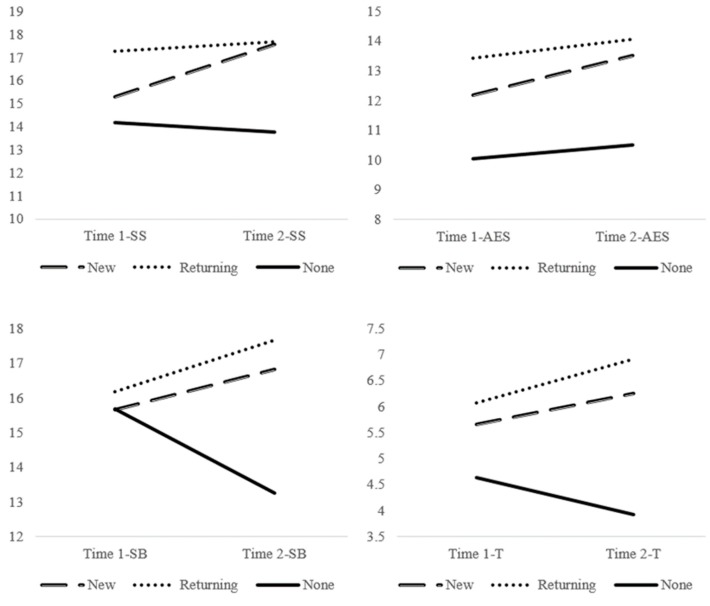
Group by time interaction shows a differential pattern of change was detected between the three intervention groups for the four GRF sub-scales; safety and structure, adult engagement and supervision, student behaviors, and transitions.

**Table 1 ijerph-17-00225-t001:** Demographic Information and Average Change Scores of Schools.

School ID	Region	Enrollment Size	% Racial/Ethnic Minority	% Economically Disadvantaged	Service Type	Average Change Scores
1	MW	322	94%	84%	New	2.67 +
2	PACNW	485	84%	95%	Returning	6.27 *
3	RM	464	95%	97%	Returning	10.67 *
4	MW	294	98%	95%	New	8.42 *
5	MW	630	47%	56%	New	12.56 *
6	NE	527	99%	71%	None	−7.39 **
7	RM	478	58%	77%	New	4.60
8	NE	457	97%	80%	New	4.46
9	RM	438	92%	93%	New	6.89 *
10	NE	635	41%	22%	Returning	−0.17
11	RM	559	88%	92%	None	5.33 *
12	PACNW	512	80%	95%	New	−5.81 **+
13	RM	428	18%	5%	Returning	−2.63
14	MW	359	91%	83%	None	−1.90
15	PACNW	513	22%	46%	Returning	5.67 *
16	RM	534	96%	95%	Returning	4.97 *
17	PACNW	518	59%	78%	New	5.67 *
18	PACNW	515	69%	95%	New	−0.86 +
19	RM	518	96%	90%	Returning	−0.42
20	PACNW	642	52%	95%	New	3.80
21	NE	941	86%	49%	New	3.92
22	NE	218	90%	80%	New	4.00
23	NE	163	83%	49%	None	0.78
24	MW	643	96%	98%	Returning	−0.33
25	NE	408	98%	65%	New	1.83 +
26	NE	525	98%	81%	None	−9.17 **

Note: * Average school score indicated meaningful change. ** Average school score indicated negative change. + School reported problems with programming and was not included in main analyses.

**Table 2 ijerph-17-00225-t002:** Descriptive Data for Time 1 and Time 2 GRF-OT and GRF-OT Subscales.

GRF-OT Score	New Playworks Schools Mean (SD)	Returning Playworks Schools Mean (SD)	Non-Intervention Schools Mean (SD)	Schools with Identified Programming Challenges Mean (SD)
Total GRF-OT Score	Time 1: 52.12 (4.54)	Time 1: 56.29 (5.88)	Time 1: 47.61 (5.40)	Time 1: 55.92 (2.96)
	Time 2: 57.63 (4.60)	Time 2: 59.85 (2.33)	Time 2: 44.52 (5.40)	Time 2: 54.48 (4.90)
Safety and Structure of Environment	Time 1: 15.32 (2.34)	Time 1: 17.26 (2.04)	Time 1: 14.19 (2.40)	Time 1: 16.31 (1.37)
	Time 2: 17.59 (1.95)	Time 2: 17.70 (1.09)	Time 2: 13.79 (1.62)	Time 2: 16.22 (1.56)
Adult Engagement and Supervision	Time 1: 12.19 (2.26)	Time 1: 13.43 (1.53)	Time 1: 10.04 (1.13)	Time 1: 12.68 (1.86)
	Time 2: 13.52 (1.37)	Time 2: 14.07 (0.89)	Time 2: 10.52 (1.45)	Time 2: 12.27 (1.86)
Student Behaviors	Time 1: 15.67 (1.55)	Time 1: 16.18 (2.04)	Time 1: 15.68 (2.02)	Time 1: 16.75 (1.20)
	Time 2: 16.84 (1.94)	Time 2: 17.66 (1.15)	Time 2: 13.24 (2.71)	Time 2: 16.23 (2.06)
Transitions	Time 1: 5.76 (1.24)	Time 1: 6.08 (1.23)	Time 1: 4.64 (1.18)	Time 1: 6.58 (0.83)
	Time 2: 6.25 (0.83)	Time 2: 6.91 (0.84)	Time 2: 3.92 (1.21)	Time 2: 6.50 (0.87)

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
