# Peer review of "Examination of the Responsiveness of the Great Recess Framework—Observational Tool"

_ijerph, 2019, doi:10.3390/ijerph17010225_

Round 1

Reviewer 1 Report

See attached file.

Author Response

Thank you for your comments. We have uploaded a point by point response. 

Reviewer 2 Report

This is an important paper given that there are very few instruments that allow researchers to investigate what happens at recess overall, rather than what individual children do. I believe you should indicate this in your Intro. The only other such measure that I know of has been developed and tested only with children with autism although it may apply more broadly. See: Grady-Dominguez, P., Bundy, A. C., Ragen, J., Wyver, S., Villeneuve, M., Naughton, G., Tranter, P., Eakman, A., Hepburn, S., and Beetham, K. (2019). An observation-based instrument to measure what children with disabilities do on the playground: A Rasch analysis. International Journal of Play. https://doi.org/10.1080/21594937.2019.1580340

Despite the disclaimer in the final paragraph of the Intro, I feel as though you have framed this manuscript more as an effectiveness study than as instrument development. Moving the content of the final paragraph up earlier in the Intro could help that; it might also lead to other changes that speak more to instrument development.

In this study, the term "recess quality" is somewhat misleading. Consider replacing it with something like "quality of structured recess". Many researchers would argue that recess should be about free play rather than activity structured by school staff. Hence what you have shown is that the GRF-OT is sensitive to change in a structured recess environment. Whether it would be sensitive to other types of change to the playground (e.g., addition of loose parts) remains to be seen.

I recommend adding to Table 1 brief information about why particular schools were eliminated from the study. That info is in the narrative but not easily accessible.

I wonder why you did not include the schools eliminated because of programmatic difficulties in the data analysis. This could add to you pool of schools not expected to change.

Please report mean and SD of Time 2 scores of each of the 3 groups as you have done for Time 1 scores. Consider a table where you report both. This gives readers a better chance to evaluate practical change.

I would also find it useful to see the sub-scores at Time 1 and Time 2 by group. You describe the changes but have not provided enough info for readers to assess the practical magnitude of the changes.

Author Response

(The authors gave the same response as above.)
